# Microcanonical Analysis of Helical Homopolymers: Exploring the Density of States and Structural Characteristics

**DOI:** 10.3390/polym15193870

**Published:** 2023-09-24

**Authors:** Matthew J. Williams

**Affiliations:** Institute of Engineering, Murray State University, Murray, KY 42071, USA; mwilliams72@murraystate.edu

**Keywords:** semiflexible polymers, conformational phases, phase transitions, helical polymers, Monte Carlo simulations, microcanonical analysis

## Abstract

This study investigates the density of states and structural characteristics of helical homopolymers. Comprising repeating identical units, the model enables the exploration of complex behaviors arising from a simple, yet generalized, set of potentials. Utilizing microcanonical analysis, transitions between helical structures are identified and categorized. Through a systematic comparison of results under varying conditions, we develop a nuanced understanding of the system’s general behavior. A two-dimensional plot illustrates the relative distribution of different structural types, effectively showcasing their prevalence. The findings of this study substantially advance our understanding of the density of states and structural transformations of helical homopolymers across a range of conditions. Additionally, the prevalence plot offers valuable insights into the occurrence of suppressed intermediate states, particularly in models featuring stiff helix segments. This research significantly enhances our understanding of the complex interactions governing helix bundling phenomena within the context of helical homopolymers.

## 1. Introduction

This study explores the use of microcanonical inflection point analysis [1,2] to understand structural transitions in the helical homopolymer model [3]. This model is a coarse-grained representation of polymer chains that exhibit helical secondary structures and helix-bundled tertiary structures. Helical structures are common in various biopolymer systems [4,5,6,7]. Finite-size effects are an essential aspect of the behavior of these systems. Although the mechanisms responsible for helix formation vary [8], we aim to study the general thermodynamic behavior in a context-independent manner.

The model used in this study consists of a set of effective potentials that coordinate the interaction of the repeating polymer units. These potentials include representations of bonded interactions, non-bonded interactions, bending angle sensitivity, and torsion angle sensitivity. Polymer structures are generated using Markov chain Monte Carlo simulation with the Metropolis–Hastings algorithm [9,10]. The efficiency of this simulation approach can be improved by including multiple parallel replicas that are permitted to exchange replicas at regular intervals [11,12,13]. In this study, the array of simulation threads is chosen to span a two-dimensional array of conditions defined by temperature and a carefully chosen model parameter that influences structure formation [14,15]. Hamiltonian exchanges are performed between threads with different model parameters [16,17,18]. Throughout the simulation, each thread produces a canonical ensemble of structures and tracks canonical averages for parameters of interest.

Although the canonical data have previously been analyzed to provide a basic understanding of structural transitions and stability [19], this paper extends that work by using histograms of canonical ensemble quantities to generate microcanonical results pertaining to each possible structure’s energy. Microcanonical analysis is used to gain deeper insight into the nature of the observed structural transitions. With the microcanonical ensemble, inflection point analysis is employed to identify and classify structural transitions [20]. Furthermore, the nature of these transitions can be better understood by considering a two-dimensional structural prevalence plot, which shows the relative prevalence of various structure types within each microcanonical ensemble [21,22,23,24].

## 2. Materials and Methods

### 2.1. Model

This study makes use of a helical homopolymer model that generates polymer structures exhibiting both secondary and tertiary structures. This model demonstrates phase transitions between random-coil, globular, and various helical structure phases. The model incorporates four potentials: the finitely extensible nonlinear elastic (FENE) potential [25], the Lennard-Jones (LJ) potential [26], the bending potential, and the torsion potential [27].

Bonded monomers interact according to the FENE potential, which is described by Equation (Equation 1). The strength of this potential depends on the distance between monomers, *r*, where a minimum potential is achieved for $r=r_{0}\equiv 1$. The bond length is not allowed to deviate from this equilibrium value by more than R≡3/7. Any move that separates two bonded monomers by more than r+R or brings them closer than r−R is immediately rejected.

Non-bonded monomers interact according to the LJ potential if they are separated by a distance of less than $r_{c}\equiv 2.5\sigma$. The LJ potential is described by Equation (Equation 2), where an energetic minimum is found for $r=r_{0}$. This is achieved with $\sigma \equiv 2^{-1/6}r_{0}$. To avoid a discontinuity at $r=r_{c}$, we add an energy shift of $v_{c}=4\left[{\left(\sigma /r_{c}\right)}^{12}-{\left(\sigma /r_{c}\right)}^{6}\right]$.

The polymer chain is additionally subject to torsion and bending potentials, which precipitate the formation of helical secondary structures. These potentials are described by Equations (Equation 3) and (Equation 4), respectively. In the case of the torsion potential, each series of three bonds has a dihedral angle τ. An energy penalty is assigned based on each dihedral angle’s variance from $\tau_{0}\equiv 0.873$. Similarly, a series of two bonds can be used to calculate their bending angle, θ. The bending energy is calculated from a bending angle’s deviation from $\theta_{0}\equiv 1.742$.
(1)$$\begin{array}{cc} v_{\mathrm{FENE}}\left(r\right) & =\log\{1-{\left[\left(r-r_{0}\right)/R\right]}^{2}\}. \end{array}$$
(2)$$\begin{array}{cc} v_{\mathrm{LJ}}\left(r\right) & =4\left[{\left(\sigma /r\right)}^{12}-{\left(\sigma /r\right)}^{6}\right]-v_{c}. \end{array}$$
(3)$$\begin{array}{cc} v_{\mathrm{bend}}\left(\theta \right) & =1-\cos(\theta -\theta_{0}). \end{array}$$
(4)$$\begin{array}{cc} v_{\mathrm{tor}}\left(\tau \right) & =1-\cos(\tau -\tau_{0}). \end{array}$$

The total energy of a polymer with configuration X can be calculated from a summation over all of the potentials as shown in Equation (Equation 5). When included in the Hamiltonian, each potential is scaled by an energy scale. These energy scales are denoted as $S_{\mathrm{FENE}}$, $S_{\mathrm{LJ}}$, $S_{\tau }$, and $S_{\theta }$.
(5)$$\begin{array}{cc} H(X)=\; & S_{\mathrm{FENE}}\sum_{i}v_{\mathrm{FENE}}\left(r_{i\;i+1}\right)+S_{\mathrm{LJ}}\sum_{i>j}v_{\mathrm{LJ}}\left(r_{ij}\right)\\ & +S_{\tau }\sum_{l}v_{\mathrm{tor}}\left(\tau_{l}\right)+S_{\theta }\sum_{k}v_{\mathrm{bend}}\left(\theta_{k}\right). \end{array}$$ The energy scale used for each potential is chosen from commonly used values in the literature [21,28]. These values are: $S_{\mathrm{FENE}}\equiv -\left(98/5\right)r_{0}^{2}R^{2}/2=-1.8$, $S_{\mathrm{LJ}}\equiv 1$, and $S_{\theta }=200$. The value of $S_{\tau }$ varies between simulation threads and influences the structure types produced by a particular simulation.

### 2.2. Two-Dimensional Parallel Tempering

The polymer model is simulated using a parallel tempering Monte Carlo approach that employs a two-dimensional array of threads with varied temperatures and Hamiltonians. This approach is used to simulate polymers comprising 30 and 40 monomers. The 30-monomer simulation consists of 160 threads, with 16 temperature values ranging from 0.2 to 1.6. These values are chosen such that the lowest temperature will settle into a distinct low-energy structure type and the highest temperature threads will produce random coil structures that easily free a replica from any local free energy minimum. Intermediate temperatures are spaced exponentially so that successive temperatures differ by the same factor. There are 10 threads with each temperature value; these threads each simulate a unique Hamiltonian with $S_{\tau }$ values between 5 and 14. The presumptive ground state structure varies as a function of $S_{\tau }$, so a single simulation is able to span several different system types. The range of values for $S_{\tau }$ is chosen so that the simulation would include single-helix systems, two-helix bundles, and intermediate systems between them. A second set of simulations is run for 40-monomer chains with a similar computational setup. In the 40-monomer simulations, $S_{\tau }$ varies between 5 and 25. Exchanges between threads *i* and *j* are accepted with a probability given in Equation (Equation 6). In this equation, $\beta_{i}$ is the inverse temperature in thread *i*, $X_{i}$ is the polymer configuration coming from thread *i*, and $H_{i}$ is the Hamiltonian for thread *i*.
(6)$$P_{\mathrm{exch}}=\min\left(1,\frac{e^{\beta_{i}H_{i}\left(X_{i}\right)}e^{\beta_{j}H_{j}\left(X_{j}\right)}}{e^{\beta_{i}H_{i}\left(X_{j}\right)}e^{\beta_{j}H_{j}\left(X_{i}\right)}}\right).$$

Each simulation thread uses a standard Metropolis algorithm with displacement updates between exchanges. Updates displace a single random monomer by a random amount. Updates are accepted with a probability $P_{\mathrm{acc}}=\min\left(1,e^{-\beta \Delta E}\right)$, where $\beta =1/(k_{B}T)$. We use units in which $k_{b}\equiv 1$. To improve simulation efficiency, three different global updates are included. For a global displacement, a monomer is chosen at random and every monomer beyond the chosen one is displaced by the same amount. To perform a bending update, a single monomer is randomly selected and all subsequent monomers rotate around an axis perpendicular to the chosen monomers neighboring bonds. Torsion update chooses a single monomer and rotates all monomers beyond the chosen monomer around an axis formed by the previous bond. The use of global updates and the 2D parallel tempering approach leads to efficient sampling of the entire range of energies for systems with a variety of tertiary structure formations.

As data are collected, each simulation thread tracks canonical ensemble average values for several measurable order parameters, most notably, energy and an order parameter, $q\equiv \sum_{|i-j|>6}\left(v_{\mathrm{LJ}}\left(r_{ij}\right)\right)/\sum_{|i-j|\leq 6}\left(v_{\mathrm{LJ}}\left(r_{ij}\right)\right)$. This order parameter measures the ratio of the total Lennard-Jones interaction between monomers separated by more than six bonds to the total Lennard-Jones interaction between monomers separated by six or fewer bonds. This parameter increases when the chain is folded to include interactions between monomers separated by many bonds. It is useful for distinguishing different helix bundling phases.

### 2.3. Multiple Histograph Reweighting

This paper focuses on a microcanonical analysis of helical polymer systems. We characterize phase transitions by analyzing the microcanonical entropy and its derivatives as functions of energy. The microcanonical entropy, defined as $S\left(E\right)=k_{B}\log\left(g\left(E\right)\right)$, is calculated by reweighting the histograms produced by individual simulation threads. A simulation thread, *i*, with temperature, *T*, produces a histogram, $h_{i}\left(E\right)$. This thread’s histogram estimates the density of states using ${\bar{g}}_{i}\left(E\right)=h\left(E\right)e^{\beta E}$. The accuracy of this estimate at a given value of *E* naturally depends on the number of counts in that energy bin.

If histograms from threads at different temperatures overlap sufficiently, they can be weighted to produce a density of states that spans all energies generated by all threads, g(E). To achieve this, we start with an initial guess for the partition function of Z=1 for all temperatures. Next, an estimate of $\hat{g}\left(E\right)$ is computed using Equation (Equation 7), where $M_{i}$ represents the total number of measurements in thread *i*. The partition function estimate can be improved using Equation (Equation 8). These equations are applied iteratively until results converge.
(7)$$\hat{g}\left(E\right)=\frac{\sum_{i}h_{i}\left(E\right)}{\sum_{i}M_{i}Z_{i}^{-1}e^{-\beta_{i}E}}.$$
(8)$$Z_{i}=\sum_{E}\hat{g}\left(E\right)e^{-\beta_{i}E}.$$ In practice, the values of g(E) are too large to work with directly, so the logarithms of g(E) and $Z_{i}$ are used instead.

From the density of states, the microcanonical entropy can be recalculated. We then employ the Savitzky–Golay technique [29] to remove noise from *S* and calculate its first three derivatives while preserving the underlying trends. The derivatives of *S*, denoted as β(E)≡dS/dE, $\gamma \left(E\right)\equiv d^{2}S/dE^{2}$, and $\delta \left(E\right)\equiv d^{3}S/dE^{3}$ are used to analyze phase transitions.

Microcanonical inflection point analysis can be used to identify phase transitions and determine their order. In the absence of phase transitions, S(E) and its derivatives will be monotonic functions that alternate between increasing and decreasing, with all derivatives asymptotically approaching zero. An *n*-th order phase transition in a finite system will be exhibited as a region of minimal sensitivity within the (n−1)-th derivative of the microcanonical entropy. This behavior is illustrated in Figure 1, where sample data are used to illustrate the inflection point or region of least sensitivity seen in the first derivative of the S(E). This inflection point corresponds to a second-order phase transition and is visible in panel (b). The second-order phase transition can also be identified by analyzing the higher-order derivatives. In panel (c), the transition manifests as a peak approaching zero and it becomes a zero-crossing in panel (d).

We can further use the density of states to calculate canonical results with higher resolution than initially sampled. The histogram of energies for a canonical ensemble with inverse temperature, β, can be calculated from the density of states using $h\left(E\right)=g\left(E\right)e^{-\beta E}$. From this histogram, the average energy can be computed $\bar{E}=\sum h\left(E\right)E\;/\sum h\left(E\right)$. By performing this calculation for an array of temperatures, we can determine the specific heat from the temperature derivative of the ensemble average energy, $C_{V}=d\bar{E}/dT$.

### 2.4. Two-Dimensional Density of States

In addition to recording the energy histogram for each thread, a two-dimensional histogram along *E* and a carefully selected order parameter are also recorded for each temperature thread. In this study, the order parameter *q* is chosen. This parameter measures the Lennard-Jones interaction between monomers separated by more than six monomers relative to the interaction with monomers less than or equal to six bonds. For the helical homopolymer system, this order parameter serves as a measure of the helix segment bundling. A two-dimensional analog of the microcanonical entropy can be calculated from these histograms using Equation (Equation 9).
(9)$$g\left(E,q\right)=\frac{\sum_{i}h_{i}\left(E,q\right)}{\sum_{i}M_{i}Z_{i}^{-1}e^{-\beta_{i}E}}.$$

Cross-sections of the density of states can be extracted from g(E,q) and scaled by g(E) to determine the relative prevalence of various structure types as a function of *E*. The structural prevalence for a particular value of *q* is calculated from p(E,q)=g(E,q)/g(E).

## 3. Results

In this section, key findings from this study are presented. Phase transitions are identified and classified by plotting the microcanonical entropy and its derivatives. For several example systems, the transition signals observed in the microcanonical analysis are compared with those in the canonical analysis. These phase transitions can be further understood by examining multiple cross-sections of a two-dimensional representation of the density of states across energy and *q*. We begin by presenting data for an array of simulations in which all polymers have a length of N=30 and will discuss polymers with N=40 in Section 3.2.

### 3.1. Microcanonical Results

In Figure 2, data are provided for three distinct system types all with length, N=30. The first column, Figure 2a,d,g,j,m, presents data pertaining to a simulation in which $S_{\tau }=5$. Here, random coil structures are observed at high energy and two-helix bundles appear at low-energy. The single-helix phase is not dominant for any energy range. The rightmost column, Figure 2c,f,i,l,o, pertains to simulations with $S_{\tau }=13$. This system generates distinctly single-helix low-energy structures. The center column, Figure 2b,e,h,k,n, represents systems with $S_{\tau }=8$. This system is intermediate to the other two, producing both single-helix and two-helix structures at low energies. Ultimately, the lowest-energy structures for this system are two-helix bundles. Further discussion on the specific makeup of the structural phases for these systems will be provided in Section 3.1.1.

In the top row of Figure 2, panels (a)–(c), the canonical ensemble specific heat is given as a function of that ensemble’s average energy, $\bar{E}$. Phase transitions typically correspond to peaks in the specific heat. A single strong peak is evident in each system shown here. Each peak represents the transition to the lowest-energy structure type. In panel (a), an additional phase transition is visible as a shoulder at a higher energy than the prominent peak. Panel (b) also exhibits a nearly hidden shoulder at a higher energy than the primary peak. These specific heat plots serve as a reference against which inflection points in the microcanonical quantities can be compared.

The microcanonical entropy, *S*, is given for three different values of $S_{\tau }$ in Figure 2d–f. The absence of inflection points in the microcanonical entropy suggests that there are no first-order phase transitions in these systems. The derivatives of the microcanonical entropy are presented in the bottom three rows, panels (g)–(o).

Figure 2g–i gives β. A region of decreased sensitivity is evident in panel (h). This region, apparent in the second derivative of *S*, corresponds to a second-order phase transition. This transition continues to be seen in the higher-order derivatives as peaks approaching zero. Additionally, this transition is observable in panel (b) as a peak in the specific heat.

All three plots of γ in Figure 2j–l feature inflection points that correspond to third-order phase transitions. In panel (j), two distinct inflection points align well with the peak and shoulder observed in the specific heat plot in panel (a). In panel (k), we see the peak associated with the second-order transition as well as an inflection point at higher energy. Again, the transitions in the microcanonical entropy plots align closely with the peak and shoulder in the specific heat. In panel (l), a single inflection point is present signifying a lone third-order transition. This transition again aligns with a transition present in the specific heat. Each of the third-order transitions, represented as regions of decreased sensitivity in the γ plots, can be clearly identified as inverted peaks in the δ plots from Figure 2m,n.

Not only are the transitions more easily identified using microcanonical analysis when compared to the canonical representation in the top row, but the transition order can be inferred from the order of derivative that first shows a region of minimal sensitivity.

A broader range of systems is explored in Figure 3. Panel (a) presents β for systems with $S_{\tau }$ values ranging from 5 to 14. Although subtle, inflection points corresponding to second-order transitions are discernible for $S_{\tau }$ of 8, 9, and 10. These transition energies are more easily identified by considering Figure 3b, which shows the second derivative of the microcanonical entropy, γ. In this figure, the three second-order transitions occur at transition energies of −19.6, −21.6, and −23.1 for the $S_{\tau }=$ 8, 9, and 10 curves, respectively.

All systems display inflection points in γ corresponding to third-order transitions with approximate transition energies of 7.0. Additionally, lower-energy third-order transitions are observed for $S_{\tau }$ values of 5, 6, 7, and 11, with transition energies of −20.8, −15.5, −16.5, and −26.2, respectively. These third-order transition energies are most easily identified by examining the inverted peaks in δ, as shown in panel (c). For $S_{\tau }=12$, a region of minimal sensitivity at E=−27.9 represents a fourth-order transition.

#### 3.1.1. Two-Dimensional Density of States

Structural prevalence is a useful way to visualize the full behavior of a system. In Figure 4, we analyze structural prevalence as a function of both *E* and order parameter *q*. Single-helix bundles have little to no LJ interaction between monomers separated by more than six bonds and therefore have a *q* value of approximately 0.01. In contrast, for two-helix bundles, each helix segment has monomers that interact with monomers from the other segment, more than six bonds away. Two-helix bundles tend to have *q* of approximately 0.35.

In Figure 4a–d, each panel gives the relative prevalence of structures across a space of *E* and *q*. A vertical slice through a single value of energy can be thought of as a normalized histogram of the structures found at each *q*. Bright yellow regions represent values of *q* which are highly represented in a single microcanonical ensemble with energy *E*. Darker regions represent more sparsely populated regions of the *q* vs. *E* space. No structures were observed within the bins that are colored white. Because the color scale is logarithmic, small changes in shade can represent rather large differences in structural prevalence. For this reason, panels (a)–(d) should be seen as more qualitative results. In order to read quantitative information from this chart, Figure 4e–h give horizontal cross-sections from each two-dimensional prevalence plot.

In all four $S_{\tau }$ examples given, the lowest-energy structures are made up entirely of structures within a single *q* bin. Figure 4a–c represents systems with $S_{\tau }$ values between 5 and 12. In all of these cases, the lowest energy structures have a value of *q* of 0.36 corresponding to two-helix bundles. As $S_{\tau }$ increases, the energetic advantage for folding decreases, and the energy of the ground state two-helix bundle approaches that of the lowest energy single-helix structures. At $S_{\tau }=14$, single-helix structures become energetically preferable to two-helix bundles. For this reason, the lowest energy populated bin in panel (d) occurs at q≈0.

As energy increases, structural variability increases as well. For the two-helix bundles, this effect is very apparent; as *E* increases the variability in *q* also increases. We see that when structures are constrained with a stronger torsion potential ($S_{\tau })$, this structure variability increase is less dramatic than in the more weakly constrained cases. For $S_{\tau }=5$, there is a secondary peak at relatively low energy and q≈0.5. This secondary peak corresponds to two-helix bundles in which the two ends of the polymer begin to wrap around each other instead of remaining constrained to their own helix segment. Within the range of $S_{\tau }$ values explored within this study, these structures are never the dominant structure type. The increase in structural variability with increasing energy is not captured as well in the single-helix case ($S_{\tau }=14$), because variability does not as quickly affect the value of *q*.

Figure 4e–h show horizontal cross-sections from the above graphs. Each curve gives the prevalence of structures of a particular value of *q* as a function of energies. In each case, the highest value of *q* given corresponds to two-helix bundles and the lowest corresponds to single-helix structures. In panels (a)–(c) we see the q=0.36 curve trend toward 1 at low energy, as all structures are represented within this single *q* bin. As energy increases, varied two-helix bundle states begin to increase in prevalence and at some point, single-helix structures also begin to increase in prevalence. The energy at which the single-helix structures increase in prevalence corresponds well to the low-energy phase transitions seen in Figure 3. Single-helix structures dominate at low energies in (d), but as energy increases we do see the presence of some two-helix bundles.

Of particular interest is the suppression of intermediate states between single-helix structures and two-helix bundles. These intermediate states are depicted by green curves in panels (e)–(h). Figure 4e makes it clear that for the third-order phase transition previously noted in the $S_{t}au=5$ system, intermediate states between the single-helix phase and the two-helix phase have significant populations. At larger values of $S_{\tau }$, the intermediate states are increasingly suppressed within the energetic window surrounding the transition from single-helix to two-helix bundle structures.

The $S_{\tau }=12$ system has a fourth-order phase transition from the single-helix phase to the two-helix bundle at E=−25. In Figure 4g, the intermediate structures are completely unrepresented as the representation of the single-helix phases decreases and that of the two-helix phase increases. At $S_{\tau }=14$, the presumptive ground state structure is a single-helix segment. The microcanonical analysis shows that there is no low-energy phase transition present in this system. However, we can observe interesting behavior with the presence of some two-helix bundles at energies slightly above those of lowest energy single-helix structures. Although they are never the dominant structure type, their presence should not be ignored. In this system, we also observe significant suppression of the intermediate states.

Figure 5 provides some example structures from the simulation with an $S_{\tau }$ of 8. Although the structures and energies differ in simulations with other values of $S_{\tau }$, they are qualitatively arranged in the same manner. Organized single-helix structures occur at low *E* and low *q*, whereas organized two-helix bundles occur at low *E* and q≈0.35. As energy increases, some structural variability is evident in the structures presented above. This variability manifests as inconsistent spacing of monomers in successive helix turns and variable spacing between helical segments. At higher energies, variability in the joint between helix segments is also found. As E≈−18, one helix segment of a two-helix bundle can be longer than the other, and at E≈20, random coil structures become dominant and structures no longer have distinct helix segments. The $S_{\tau }=8$ system was chosen because there is a significant representation of the intermediate structures falling in between the single-helix phase and the two-helix bundle phase. As $S_{\tau }$ increases, this intermediate region becomes so sparse that a simple time series of structures from all canonical ensembles fails to capture a representative sample.

### 3.2. Alternate System Size

In addition to 30-monomer systems, the same approach has been applied to a system with 40 monomers. Figure 6 presents the one-dimensional microcanonical quantities for the 40-monomer helical homopolymer model. The range of $S_{\tau }$ values is adjusted to stabilize the same tertiary structures observed in the 30-monomer results. At $S_{\tau }=8.3$, two-helix bundles are the sole low-energy structure type produced and at $S_{\tau }=25$, single-helix structures dominate at low energies. Intermediate values of $S_{\tau }$ are also presented.

We can see in Figure 6a that systems with $S_{\tau }$ of 8.3, 11.7, 15, and 18.3 all exhibit second-order phase transitions at energies of 3.0, −10.4, −18.8, and −27.5, respectively. These correspond to the transition between low-energy two-helix bundles and a single-helix phase, which dominates at an intermediate energy range. For Sτ=21.7, the transition between two-helix bundles and single helices is a third-order transition occurring at an energy of −34.9.

Systems with $S_{\tau }$ values of 18.3, 21.7, and 25.0 clearly exhibit a third-order transition at E≈10. This corresponds to a transition between random-coil structures and more organized single-helix structures. Although this transition is present in all of the systems studied here, the signal is obscured by the peak corresponding to lower-order transitions seen in the $S_{\tau }=8.3$, 11.7, and 15.0 cases. Evidence of this transition is most easily observed as an inverted peak in the δ plot from panel (c).

Figure 7 presents a two-dimensional prevalence representation for the N=40 systems. Qualitatively, the behavior of the 40-monomer system closely resembles that of the N=30. At $S_{\tau }=8.3$, there is a strong low-energy representation of two-helix bundle structures at q≈0.35. With a slight increase in energy, some structures emerge with a *q* value of approximately 0.6; this peak corresponds to three-helix bundles. These structures are poorly stabilized and form with several different orientations. Although there are single-helix structures formed for $S_{\tau }=8.3$, they are only represented in the relatively higher energy range of E>−30.

For systems with $S_{\tau }$ greater than 8.3, several changes become apparent. At $S_{\tau }=15$ the presence of three-helix bundles completely disappears. The lowest energy for the single-helix phase decreases and eventually approaches that of the two-helix bundles in the $S_{\tau }=21.7$ case. As the increased $S_{\tau }$ brings the energies of the two-helix and single-helix phases closer together, intermediate structures become increasingly entropically suppressed. This effect is evident at $S_{\tau }=15$ when considering the green curves in panel (f). The $S_{\tau }=21.7$ system demonstrates this effect even more strongly, with intermediate phases nearly entirely unrepresented in our simulation. The low-energy dominance of the single-helix structures is apparent in the $S_{\tau }=25$ system. In panels (d) and (h), very few two-helix bundle structures are observed.

## 4. Discussion

Microcanonical inflection point analysis is gaining wider acceptance in the realm of computational statistical physics, particularly for its ability to provide a nuanced understanding of phase transitions [30,31,32,33]. In our study, this method has proven useful for identifying, classifying, and understanding structural transitions in helical homopolymers. The flexibility of the approach means that it can be applied wherever the density of states is known, suggesting its utility across a broad range of disciplines including condensed matter physics and biological systems. Additionally, we find that incorporating an order parameter to differentiate phases allows us to produce a two-dimensional prevalence plot, which offers additional insights into the physical behavior of the system under study.

In this paper, we expand on earlier research on the helical homopolymer model, where phase transitions were identified and a hyperphase diagram was constructed through analysis of canonical quantities [28]. We employ microcanonical inflection point analysis to gain a deeper understanding of phase transitions in polymers with lengths of N=30 and N=40. Specifically, we are able to more precisely measure phase transition energies and determine the order of each transition. The energy scale for torsion potential, $S_{\tau }$, is tuned to achieve varying helical segment stiffness [3]. Across all systems investigated in this paper, we find a third-order phase transition between single-helix structures and random coil structures. In cases where there is an energetic preference for two-helix bundles, an additional transition from single-helix to two-helix bundles occurs at lower energy. The order of this transition is influenced by the stiffness of the helix: stiffer systems show second-order transitions, whereas more weakly confined systems exhibit third-order transitions. Despite the variations in transition energy and helix stiffness, the order of phase transitions remains consistent across both system sizes examined in this study.

Utilizing two-dimensional prevalence plots, we observe varying degrees of entropic suppression for intermediate states between the two-helix and single-helix phases. Generally, we find that systems with stiffer helical segments exhibit greater entropic suppression of these intermediate structures. This trend holds true for both the 30-monomer and the 40-monomer systems. The entropic suppression of these intermediate states stems from their significantly increased energy levels in models with stiff helical segments. From a practical standpoint, this entropic suppression complicates accurate sampling when using traditional one-dimensional replica exchange schemes. In our case, incorporating a two-dimensional replica exchange scheme greatly improved simulation efficiency.

## 5. Conclusions

This study underscores the utility of microcanonical inflection point analysis for studying complex systems. We used this method to analyze phase transitions in the helical homopolymer model. We found consistent third-order transitions between single-helix and random coil structures across different system sizes. Additionally, systems with an energetic preference for two-helix bundles exhibited an extra transition, the order of which varied based on helix stiffness. The methodology proved to be a robust tool for understanding phase transitions, thereby broadening its applicability across statistical systems.

There are several intriguing avenues for future work related to this publication; although the utility of microcanonical inflection point analysis is evident, this technique could benefit from further formalization and refinement to promote its broader adoption across various disciplines. With respect to the helical homopolymer model, conducting a microcanonical analysis of longer chains would permit a more comprehensive investigation of transitions in systems featuring stable three-helix bundle. 

## Figures and Tables

**Figure 1 polymers-15-03870-f001:**
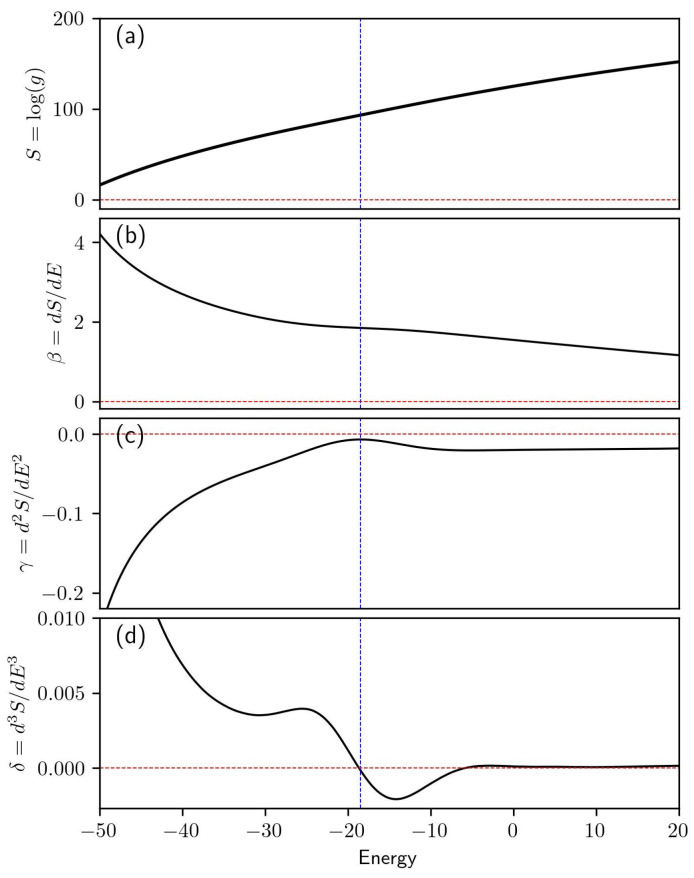
Illustration of example data for microcanonical inflection point analysis. The vertical blue line marks the transition energy for a second-order phase transition. The entropy *S* is provided in panel (**a**). Panels (**b**–**d**) give the first, second, and third derivatives of the entropy, respectively.

**Figure 2 polymers-15-03870-f002:**
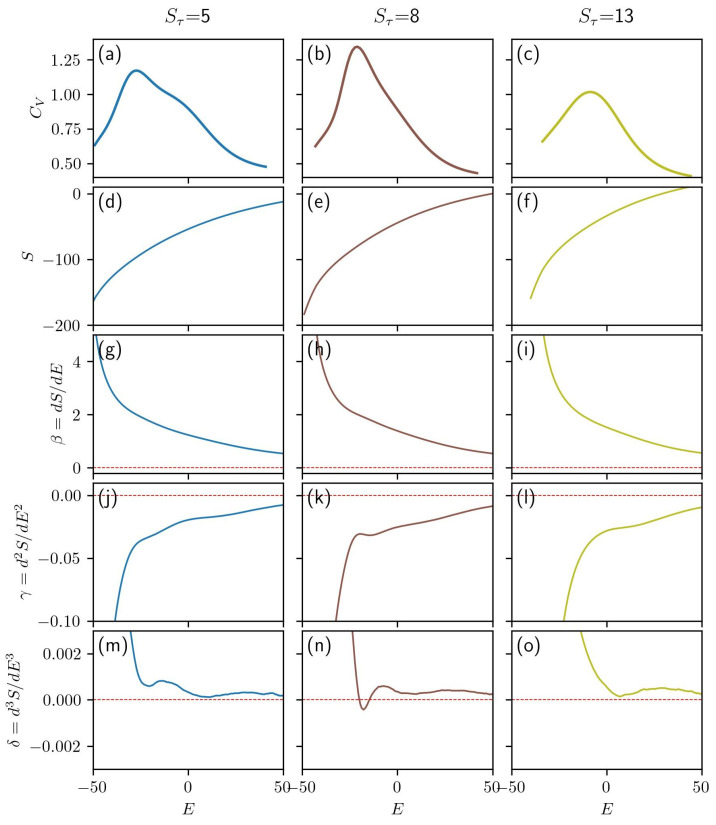
(**a**–**c**) For three different values of $S_{\tau }$, the canonical ensemble specific heat is given as a function of each ensemble’s average energy. The microcanonical entropy is given in (**d**–**f**). The first (**g**–**i**), second (**j**–**l**), and third (**m**–**o**) derivatives of microcanonical entropy are given as a function of energy. All panels in this figure represent polymers of length N=30.

**Figure 3 polymers-15-03870-f003:**
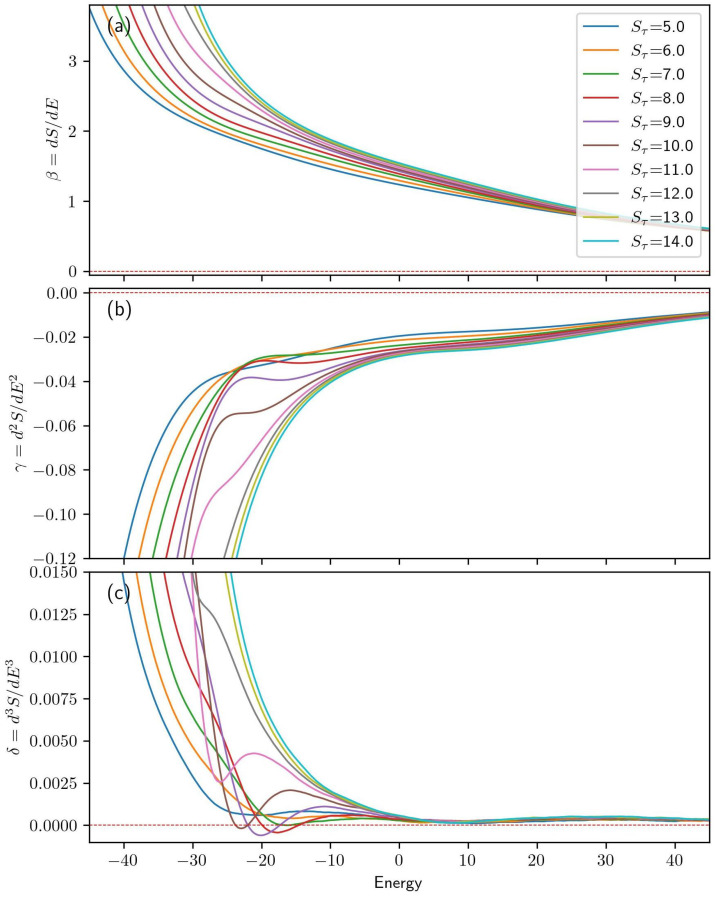
(**a**) For each value of $S_{\tau }$ in the N=30 simulation, the first derivative of microcanonical entropy is given as a function of energy. Panels (**b**,**c**) present the second and third derivatives of the microcanonical entropy, respectively.

**Figure 4 polymers-15-03870-f004:**
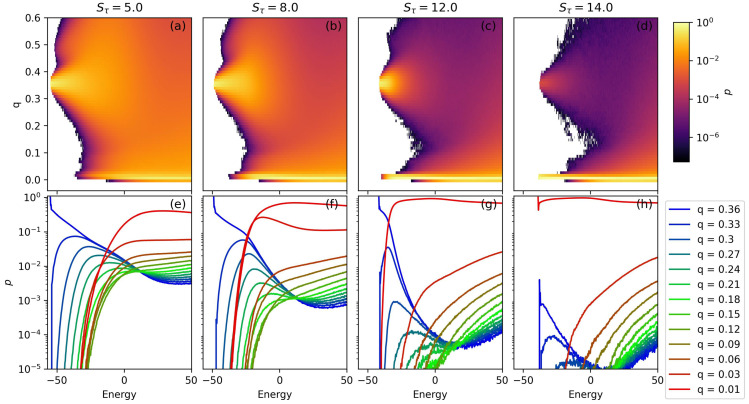
(**a**–**d**) A two-dimensional structural prevalence plot is given for several interesting values of $S_{t}au$ in the N=30 simulation. In these figures, the structural prevalence is given as a function of structure energy and order parameter, *q*. Low-energy two-helix bundles have q≈0.35 and single-helix structures have q≈0.01. Below, in (**e**–**h**), cross-sections of the two-dimensional structural prevalence plot are given. Values of *q* are chosen such that p(E) is shown for two-helix bundles, single-helix structures, and an array of intermediate structures.

**Figure 5 polymers-15-03870-f005:**
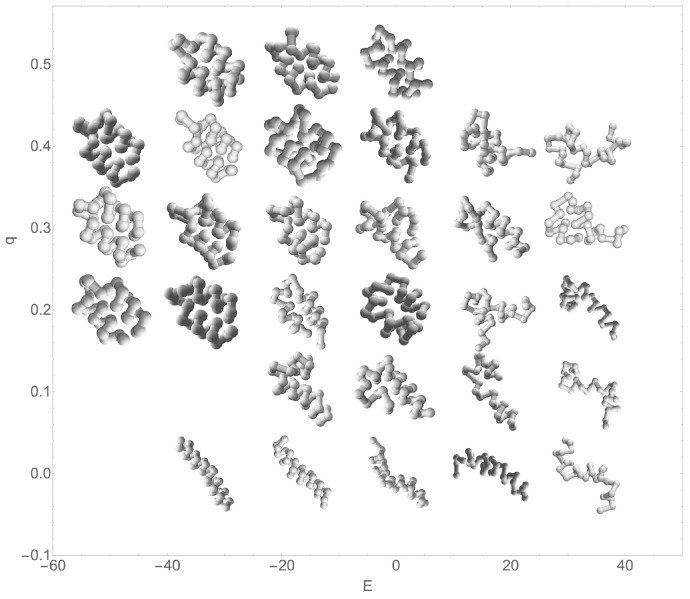
Example structures from the $S_{\tau }=8$ simulation. Structures are plotted in the approximate location of their *E* and *q*.

**Figure 6 polymers-15-03870-f006:**
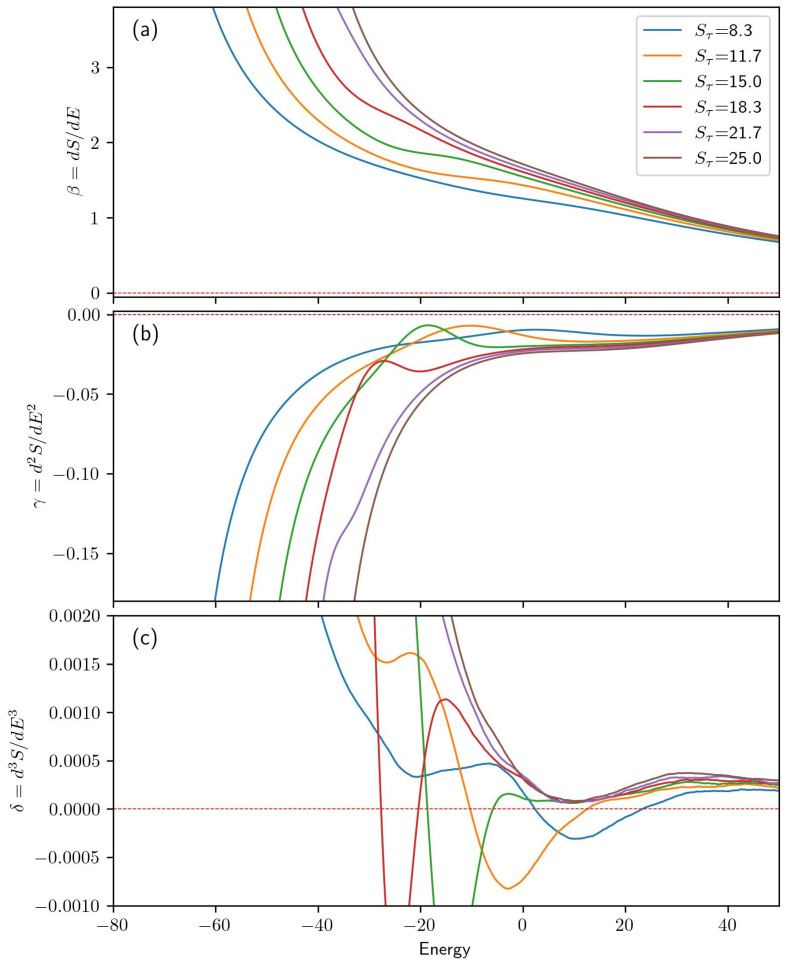
(**a**) For each value of $S_{\tau }$ in the N=40 simulation, the first derivative of microcanonical entropy is given as a function of energy. (**b**,**c**) give the second and third derivatives of the microcanonical entropy, respectively.

**Figure 7 polymers-15-03870-f007:**
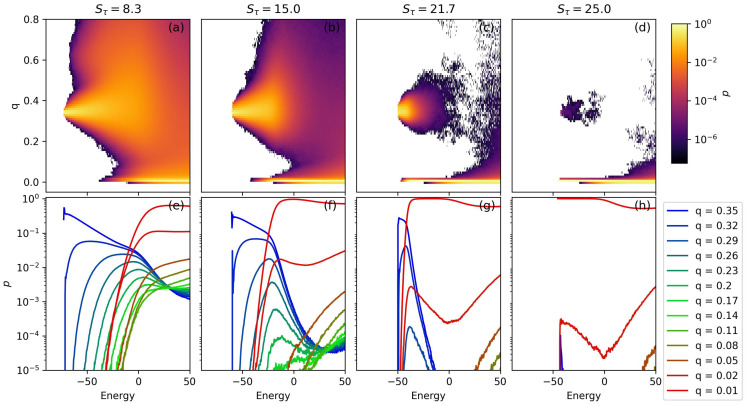
(**a**–**d**) A two-dimensional structural prevalence plot is given for several interesting values of $S_{t}au$ in the N=40 simulation. In these figures, the structural prevalence is given as a function of structure energy and order parameter, *q* and *E*. Low-energy two-helix bundles have q≈0.35 and single-helix structures have q≈0.01. Below, in (**e**–**h**), cross-sections of the two-dimensional structural prevalence plot are given. Values of *q* are chosen such that p(E) is shown for two-helix bundles, single-helix structures, and an array of intermediate structures.

## Data Availability

The data that support the findings of this study are available from the corresponding author upon reasonable request.

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
