# Peer review of "Microcanonical Analysis of Helical Homopolymers: Exploring the Density of States and Structural Characteristics"

_polymers, 2023, doi:10.3390/polym15193870_

Round 1

Reviewer 1 Report

1.         Relevant research background needs to be supplemented in INTRODUCTION.

2.         The CONCLUSIONS is not found in this paper and should be added.

3.         The conclusion “the order of this transition depends on the helix stiffness” is lack of sufficient explanation.

4.         The significance of this paper is not expounded sufficiently. The author needs to highlight this paper's innovative contributions.

5.         Please do not use various fonts in the figures. The labels should be of the same font and size throughout the figures.

1.         The manuscript needs careful editing and particular attention to punctuation and spelling. There is at least one sentence without punctuation in the manuscript, such as, in line 57 and 89. And in line 179, “(a), (d), (g), (h), and (m)” would be “(a), (d), (g), (j), and (m)”.

Author Response

Dear Reviewer 1,

I would like to express my gratitude for the time and effort you have invested in reviewing my manuscript. The constructive feedback has significantly improved the quality of the paper.

I have carefully addressed all the comments and suggestions. Below is a summary of the major revisions I've made based on your comments as well as the comments of the other Reviewer:

1. External assistance was enlisted to improve the spelling and grammar throughout the manuscript.
2. The introduction has been expanded to include additional supplementary information and citations, providing a more comprehensive background on the topic.
3. The unique contributions of this study to the field are now more explicitly stated in both the introduction and the discussion sections.
4. Relevant citations have been added to the discussion section to support the paper's claims and situate them within the existing literature.
5. I have clarified the statement, "the order of this transition depends on the helix stiffness," to eliminate any ambiguity.
6. Graphics have been revised to maintain a consistent font size across the paper.
7. Discrepancies in figure panel references have been corrected to align with the included versions of the figures.
8. Explanations have been added to detail the rationale for the selected energy values.
9. Scales for the x- and y-axes have been included in Figures 1 and 2.
10. In response to Reviewer 1, a conclusion section has been added to the paper.

Thank you for your time and consideration. I look forward to your feedback on these revisions.

Sincerely,

Matthew J Williams

Reviewer 2 Report

The article by Matthew J. Williams presents a well-designed computational study on helical homopolymers. Based on Markov chain Monte Carlo simulations, various helical structures are identified, compared and classified. The study yields a comprehensive image on the landscape of conformations for helix-forming polymer chains. Its outcome is of general value for polymer chemists dealing with helical structures.

 In general, the article is well written, argumentations are clear at any point. The only general weak spot may be the discussion (section 4). It is relatively short and without any reference to previous work. What I am missing at this point would be references to other approaches of the same kind (MD simulations of polymer chains) or maybe experimental measurements that could support the simulated results. Even comparison to own work like [2] and [3] would be appropriate. The article would gain a lot of relevance with some additions at this point.

In addition, I would see some minor issues that should be taken care of:

 1)      At the end of section 2.1, the author proposes scaling values for the different energies. Are these just random numbers, or is there a justification for these settings? A short comment at this point would be helpful.

2)      In Fig. 1, the scaling of the energy values and of the third derivative in (d) is undefined: only the zero line is marked for Delta, and no numbers are found along the energy axis.

3)      In Fig. 2, the energy axis is completely unlabeled.

4)      In the captions to Figs. 4 and 7: I guess it should read (e) - (h) instead of (e) - (f).

5)      There is a leftover bracket at the end of the keyword list.

With the changes mentioned above, I would regard the article as suitable for publication in POLYMERS.

Author Response

Dear Reviewer 2,

I would like to express my gratitude for the time and effort you have invested in reviewing my manuscript. The constructive feedback has significantly improved the quality of the paper.

I have carefully addressed all the comments and suggestions. Below is a summary of the major revisions I've made based on your comments as well as the comments of the other Reviewer:

1. External assistance was enlisted to improve the spelling and grammar throughout the manuscript.
2. The introduction has been expanded to include additional supplementary information and citations, providing a more comprehensive background on the topic.
3. The unique contributions of this study to the field are now more explicitly stated in both the introduction and the discussion sections.
4. Relevant citations have been added to the discussion section to support the paper's claims and situate them within the existing literature.
5. I have clarified the statement, "the order of this transition depends on the helix stiffness," to eliminate any ambiguity.
6. Graphics have been revised to maintain a consistent font size across the paper.
7. Discrepancies in figure panel references have been corrected to align with the included versions of the figures.
8. Explanations have been added to detail the rationale for the selected energy values.
9. Scales for the x- and y-axes have been included in Figures 1 and 2.
10. In response to Reviewer 1, a conclusion section has been added to the paper.

Thank you for your time and consideration. I look forward to your feedback on these revisions.

Sincerely,

Matthew J Williams

Round 2

Reviewer 1 Report

The authors have revised the manuscript according to the reviewers' comments.